 

# The association between BMI and serum uric acid is partially mediated by gut microbiota

Zhuo Duan,[1] Jingxiang Fu,[1] Feng Zhang,[1] Yijia Cai,[1] Guangyan Wu,[1] Wenjun Ma,[2] Hongwei Zhou,[1] Yan He[1,3]

**ABSTRACT**    Obesity is a risk factor for the development of hyperuricemia, both of which were related to gut microbiota. However, whether alterations in the gut microbiota lie in the pathways mediating obesity's effects on hyperuricemia is less clear. Body mass index (BMI) and serum uric acid (SUA) were separately important indicators of obesity and hyperuricemia. Our study aims to investigate whether BMI-related gut microbiota characteristics would mediate the association between BMI and SUA levels. A total of 6,280 participants from Guangdong Gut Microbiome Project were included in this study. Stool samples were collected for 16S rRNA gene sequencing. The results revealed that BMI was significantly and positively associated with SUA. Meanwhile, BMI was significantly associated with the abundance of 102 gut microbial genera, 16 of which were also significantly associated with SUA. The mediation analysis revealed that the association between BMI and SUA was partially mediated by the abundance of *Proteobacteria* (proportion mediated: 0.94%, *P* < 0.05). At the genus level, 25 bacterial genera, including *Ralstonia, Oscillospira, Faecalibacterium*, etc., could also partially mediate the association of BMI with SUA (the highest proportion is mediated by *Ralstonia*, proportion mediated: 2.76%, *P* < 0.05). This study provided evidence for the associations among BMI, gut microbiota, and SUA, and the mediation analysis suggested that the association of BMI with SUA was partially mediated by the gut microbiota.

**IMPORTANCE**    Using 16S rRNA sequencing analysis, local interpretable machine learning technique analysis and mediation analysis were used to explore the association between BMI with SUA, and the mediating effects of gut microbial dysbiosis in the association were investigated.

**KEYWORDS**    body mass index, 16S rRNA, microbiota, serum uric acid

Hyperuricemia (HUA) is a metabolic disease characterized by an elevated level of serum uric acid (SUA) (1). It was reported that the prevalence of hyperuricemia in the Chinese adult population increased from 11.1% in 2015–2016 to 14.0% in 2018–2019 (2). Accumulating data have revealed that hyperuricemia is a crucial risk factor for gout, hypertension, cardiovascular, diabetes, atherosclerosis, and chronic kidney disease (3–8). Generally, hyperuricemia is attributed to urate overproduction in the liver and insufficient urate elimination in the renal or extra-renal (9). Obesity has also been reported as a risk factor for hyperuricemia (10). As early as 2004, a study of 17,155 children in Japan found that the detection rate of hyperuricemia was 2.5% in normal-weight children, while it was up to 12% in obese children (11). Recently, a cross-sectional study of 144,856 adults showed that overweight people were more likely to develop hyperuricemia (12).

The biological mechanisms of obesity-induced hyperuricemia have been explored extensively in both experimental and epidemiological studies, including the excess ingestion of purine-rich foods, impaired renal clearance of uric acid, and increased production and secretion of uric acid by adipose tissue (10, 13, 14). In addition, lactate

Address correspondence to Yan He, yanhe@i.smu.edu.cn.

Zhuo Duan, Jingxiang Fu, and Feng Zhang contributed equally to this article. Author order was determined by drawing straws.

The authors declare no conflict of interest.

accumulation in obesity accelerates renal urate reabsorption via urate transporter 1 (URAT1), and the obesity gene product leptin can affect the renal clearance of uric acid (15, 16). Recent studies have shown that hyperuricema may also be associated with alterations in the gut microbiome (17). Specifically, animal studies have demonstrated that hyperuricemia mice exhibit changes in the diversity and abundance of gut microbiota, which potentially affects amino acid metabolism and promotes purine metabolism disorder and inflammation (18, 19). Similarly, individuals with hyperuricemia have been found to have decreased microbial diversity and an altered gut microbiota composition compared to those with normouricemia (17). In children with hyperuricemia, serum uric acid level was positively correlated with genera *Actinomyces*, *Morganella*, and *Streptococcus*, and negatively associated with the short-chain fatty acids-producing bacteria, such as *Alistipes*, *Faecalibacterium,* and *Oscillospira*, and sulfur-producing bacteria *Bilophila* (20). Previous studies have shown an association between SUA, gut microbiota, and obesity. For example, Gong et al. demonstrated that the counts of *Bifidobacterium* significantly decreased in visceral obesity, and serum uric acid may be a mediator between the reduction of *Bifidobacterium* and the increase of visceral adipose tissue (21). These findings imply that the alterations in the gut microbiota components might partially mediate the association between BMI with SUA.

Whether some gut microbial features lie in the pathways from obesity to hyperuricemia is less clear. In the current study, we investigated the association between BMI and SUA in a community-based population and further investigated the mediating effects of gut microbial dysbiosis in the association. Our findings provided important clues to further elucidate the pathogenesis of elevated SUA levels and aid in the prevention of hyperuricemia resulting from increased BMI or obesity.

## MATERIALS AND METHODS

### Study design and participants

The Guangdong Gut Microbiome Project (GGMP) was a community-based cross-sectional study cohort conducted in 14 districts in Guangdong Province, China. Multistage cluster sampling method was used to select the population from 14 surveillance sites for the survey. All participants must be at least 18 years of age and signed an informed letter. Detailed information regarding study design could be found in our previous research (22). After removing the samples obtained from those who did not have BMI and UA values in the metadata and those with stool sample sequences of less than 10,000 reads, a total of 6,280 samples were retained in this study. We performed an analysis of the missing rate for the host baseline characteristics (Fig. S1). Those characteristics with more than 5% missing were excluded, and the remaining with less than 5% missing were imputed using random forest algorithm. A comparison of baseline population characteristics before and after imputing missing data showed that the differences were not significant, suggesting that the missing data might be missing randomly (Table S1).

### Biological sample collection and testing

All participants had a face-to-face interview followed by a medical examination and sample collection by medical professionals. Information on anthropometrics, physiological and biochemical indicators, dietary habits together with the use of drugs and antibiotics was recorded. Fresh peripheral blood samples and stool samples were collected from all participants and sent to the laboratory authorized by Guangdong Center for Disease Control and Prevention through cold chain transport. In our previous study, we described the method for processing the samples (22). In short, we extracted total bacterial DNA from stool samples. The barcode primers (5′ to 3′) used to amplify the V4 region of the 16S rRNA gene were V4F, GTGYCAGCMGCCGCGGTAA and V4R, GGACTACNVGGGTWTCTAAT. PCR amplification conditions were as follows: initial denaturation at 94°C for 5 minutes, followed by 30 cycles of denaturation at 94°C for

30 seconds, annealing at 52°C for 30 seconds, extension at 72°C for 45 seconds, and final extension at 72°C for 5 minutes. Next-generation sequencing of the PCR products was performed on an Illumina HiSeq 2500 platform at the Beijing Genome Institute (BGI; Beijing, China).

## Definition of key measures

Hyperuricemia was defined as SUA >420 µmol/L for men, >360 µmol/L for women based on diagnostic criteria (23). Here, we employed four operational taxonomic unit (OTU)-based α-diversity indices: Chao1, Shannon, observed OTUs, and phylogenic diversity (PD) whole tree. And assessed four bacteria (*Bacteroidetes*, *Proteobacteria*, *Firmicutes*, and *Verrucomicrobia*) at the phylum level due to the high abundance (> 92%) observed in our previous study (22).

## Bioinformatics and biostatistics

The procedure for pre-processing the raw sequences was described in our earlier paper (22). In brief, QIIME V.1.9.1 was used to merge the data, perform quality control assessments, and analyze the data at the sequence level with the non-reference statistical denoising deblur method to avoid misidentification of different operational taxonomic unit-based data derived from misclustered sequences (24). Representative sequences were aligned using the PyNAST algorithm (25). The Ribosomal Database Project classifier against the Greengenes database (version 13.8) was used to perform taxonomic profiling (26, 27).

The missing values of the host characteristic were analyzed and visualized in this study with the R package "naniar" (version 0.6.1) and imputed with the R package "missForest" (version 1.5) (28). The completed metadata data set was then divided into a 70% training set, 20% validation set, and 10% unseen test to establish the random forest classification model, which could distinguish between healthy participants and those with high serum uric acid. In addition, the categorical variables used in this model were coded by One-Hot encoding method and the target class-imbalance fixed by Synthetic Minority Oversampling TEchnique (SMOTE). SHapley Additive exPlanations (SHAP) analysis was adapted to better assess the model's feature importance. Briefly, SHAP is a game theoretic approach that combines optimal credit allocation with local explanations by using classical Shapley values to explain the output of the machine learning model (29). The Python library "Pycaret" (version 2.3.10) was used to perform the above process and to optimize the hyperparameters of the random forest model.

In the microbiome mediation analysis section, we first excluded the data at genus level with a prevalence under 5% to reduce statistical noise from low-prevalence microbiota. Using the cleaned data, Spearman correlations were calculated between BMI and genus-level microbes, and significantly associated microbes were examined using a threshold of False Discovery Rate (FDR) <0.1. The R package "bruceR" (version 0.8.9) was then used to calculate whether microbiota played a mediating role between BMI and UA and its value. In brief, a mediator formula (microbiota = covariates + BMI) and an outcome formula (UA = covariates + BMI +microbiota) were created to calculate the direct effect, indirect effect, and the total effect. The mediated proportion was calculated by dividing indirect effect by total effect. All analyses were performed on the 6,280 samples and the top 15 baseline characteristics associated with risks of HUA were chosen as covariates in all correlation and mediation analyses: age, gender, triglyceride (TG), blood urea nitrogen (BUN), high density lipoprotein (HDL), hemoglobin (Hb), low density lipoprotein (LDL), fasting blood glucose (FBG), systolic blood pressure (SBP), diastolic blood pressure (DBP), alaninetransaminase (ALT), total cholesterol (TCHO), grains, livestock_meat, and fruits.

## RESULTS

### Characteristics of study participants

The study population was previously described in the GGMP, which contains 7,009 individuals from 14 districts within Guangdong Province, China. Participants with missing information on BMI and uric acid were excluded. Finally, 6,280 participants remained for subsequent analysis.

The participants were divided into two groups: the normal (Nor) group and the hyperuricemia (HUA) group (Table 1). The number of participants with HUA were 1,494 (23.78%) with a median SUA level of 445 µmol/L (IQR: 411–490) and a median BMI level of 24.6 kg/m$^2$ (IQR: 22.3–26.9). Most of the clinical indicators are statistically different between the two groups. Compared with Nor group, those with HUA had higher levels of SBP, DBP, FBG, TCHO, TG, LDL, Hb, ALT, and BUN, but lower levels of HDL (Table 1).

### Associations between BMI and hyperuricemia using local interpretable machine learning technique

We build a random forest classifier model with the clinical characteristics shown in Table 1 to evaluate their contribution on distinguishing subjects with hyperuricemia from healthy counterparts, which helped us to discover the associations between characteristics and hyperuricemia. By using the built-in random forest's Gini impurity method, we found that BMI was the second most contributed feature among all characteristics (Fig. S2). Though the traditional global explanation approach, such as mean decrease in Gini impurity used in the random forest model, is widely applied and easy to use, it could only list out the feature contribution level in the model and could not offer the direction of the feature's impact on the model. However, the local interpretable machine learning technique such as SHAP could allow us to supplement the deficiencies mentioned above. In Fig. 1A, the characteristics were ranked based on their mean absolute SHAP values, which were in line with the global importance of the feature, and each point indicated a value for a subject. Meanwhile, those points locate on the positive side of the horizontal axis, which means they have a higher probability of hyperuricemia, and the other points that lay on the negative side of the axis represent a lower probability of hyperuricemia. Combined with the color of the dot, which shows the original value of the feature, we could simply find that the one with a higher BMI seems to have a higher probability of hyperuricemia. It is noteworthy that the SHAP technique helps us to take an insight into the relationship between BMI and hyperuricemia. Individuals with a BMI above 24 are more likely to develop hyperuricemia than those who are thinner (Fig. 1B).

### Association of BMI with the gut microbiota

There was no significant correlation between BMI and α-diversity of gut microbiota. However, among the four bacteria at the phylum level (*Bacteroidetes*, *Proteobacteria*, *Firmicutes*, and *Verrucomicrobia*), the abundance of *Proteobacteria* and *Firmicutes* increased and that of *Verrucomicrobia* and *Bacteroidetes* decreased with increasing BMI (Table 2). We also found 102 genera that were significantly related to BMI (FDR 0.1). Among them, the abundance of 50 genera showed a decreasing trend with increasing BMI, while the abundance of other species showed an increasing trend (Table S2). For example, *Blautia* and *Ralstonia* were positively associated with BMI. In contrast, *Parabacteroides* species, which may help maintain intestinal homeostasis, were inversely associated with BMI. In addition, some butyric acid-producing bacteria, such as *Butyricimonas* and *Oscillospira*, were also negatively associated with BMI (Table S2).

### Association of gut microbiota and SUA

The results of the association between gut microbiota and SUA revealed that Chao1, one of α-diversity indices, was negatively associated with SUA (β = −0.158, 95% CI: −0.270, −0.045, *P* = 0.006) (Fig. 2A). In the analyses of the four phylum levels, increased abundance

**TABLE 1** General characteristics of study participants[a]

| Characteristics | HUA (N = 1,494) | Nor (N = 4,786) | P-value |
|---|---|---|---|
| BMI, median (IQR) | 24.6 (22.3–26.9) | 22.6 (20.5–24.9) | <0.001 |
| UA (µmol/L), median (IQR) | 445 (411–490) | 298 (256–342) | <0.001 |
| Age, median (IQR) | 56 (45–65) | 53 (43–63) | <0.001 |
| Gender (female), n (%) | 654 (44) | 2,834 (59) | <0.001 |
| Education, n (%) | | | 0.001 |
| Middle school or lower | 1,082 (72) | 3,684 (76.9) | |
| High school or professional college | 238 (15.9) | 679 (14.1) | |
| University | 174 (11.6) | 423 (8.8) | |
| Smoke now, n (%) | 435 (29) | 1,173 (25) | <0.001 |
| Sleep_time (min), median (IQR) | 480 (420–480) | 480 (420–510) | 0.03 |
| SBP (mmHg), median (IQR) | 133 (121–148) | 126 (115–142) | <0.001 |
| DBP (mmHg), median (IQR) | 79 (72–87) | 76 (69–83) | <0.001 |
| [b]HR (/min), median (IQR) | 76 (70–84) | 76 (69–83) | 0.603 |
| FBG (mmol/L), median (IQR) | 5.43 (5.03–5.98) | 5.27 (4.88–5.74) | <0.001 |
| TCHO (mmol/L), median (IQR) | 5.33 (4.82–5.86) | 5.18 (4.68–5.72) | <0.001 |
| TG (mmol/L), median (IQR) | 1.42 (1.01–2.14) | 0.98 (0.72–1.44) | <0.001 |
| HDL (mmol/L), median (IQR) | 1.12 (0.92–1.37) | 1.26 (1.04–1.51) | <0.001 |
| LDL (mmol/L), median (IQR) | 3.40 (2.81–4.07) | 3.12 (2.56–3.74) | <0.001 |
| Hb (g/L), median (IQR) | 148 (134–159) | 142 (130–153) | <0.001 |
| ALT (U/L), median (IQR) | 17 (12–25) | 15 (11–21) | <0.001 |
| BUN (mmol/L), median (IQR) | 5.39 (4.55–6.49) | 4.99 (4.17–5.95) | <0.001 |
| Gout, n (%) | 97 (6.5) | 109 (2.3) | <0.001 |
| [c]Bristol stool type, n (%) | | | 0.025 |
| 1&2 | 96 (6.4) | 440 (9.1) | |
| 3&4 | 1,154 (77.2) | 3,624 (75.7) | |
| 5&6 | 236 (15.7) | 704 (14.7) | |
| 7 | 8 (0.5) | 18 (0.3) | |
| Synbiotics use within 1 month, n (%) | 47 (3.2) | 121 (2.6) | 0.198 |
| Antibiotics use within 1 month, n (%) | 86 (5.9) | 289 (6.2) | 0.707 |
| Drug use within 3 days, n (%) | 295 (20) | 764 (16) | <0.001 |
| Grains (g), median (IQR) | 106,412 (65,700–160,052) | 109,500 (73,000–164,250) | 0.003 |
| Vegetables (g), median (IQR) | 109,500 (73,000–164,250) | 109,500 (73,000–164,250) | 0.725 |
| Fruits (g), median (IQR) | 15,600 (5,200–39,000) | 15,600 (4,320–36,500) | 0.005 |
| Livestock meat (g), median (IQR) | 36,500 (17,238–73,000) | 36,500 (16,900–70,905) | 0.063 |

[a]Kruskal-Wallis rank sum test for continuous variables and Pearson's chi-squared test for categorical variables.
[b]HR, heart rate.
[c]Types 1 and 2 describe stool that is hard to pass and may point to constipation. Types 3 and 4 describe stool that is well-formed and easy to pass. Types 5 and 6 may point to loose stools. Type 7 describes very loose stools or fully liquid diarrhea.

of *Proteobacteria* was associated with higher SUA (β = 35.152, 95% CI: 11.447, 58.858, *P* = 0.004), whereas *Bacteroidetes*, *Firmicutes*, and *Verrucomicrobia* were not significantly associated with SUA (Fig. 2B). Next, we only chose BMI-associated genera in the analysis of the association between SUA and gut microbes. We found 16 genera were also significantly associated with SUA (*P* < 0.05). Especially, *Odoribacter* and *Pyramidobacter* were negatively associated with BMI and SUA, and *Megamonas*, *Clostridium*, *Acinetobacter*, *Chryseobacterium, Novosphingobium*, etc., were positively associated with BMI and SUA (Table S3).

## The mediation effects of the gut microbes on the association between BMI and SUA

Mediation analysis indicated that the associations of BMI with SUA were partially mediated by gut microbiota. For example, at the phylum level, the relative abundance of *Proteobacteria* was positively associated with higher BMI and SUA (proportion mediated:

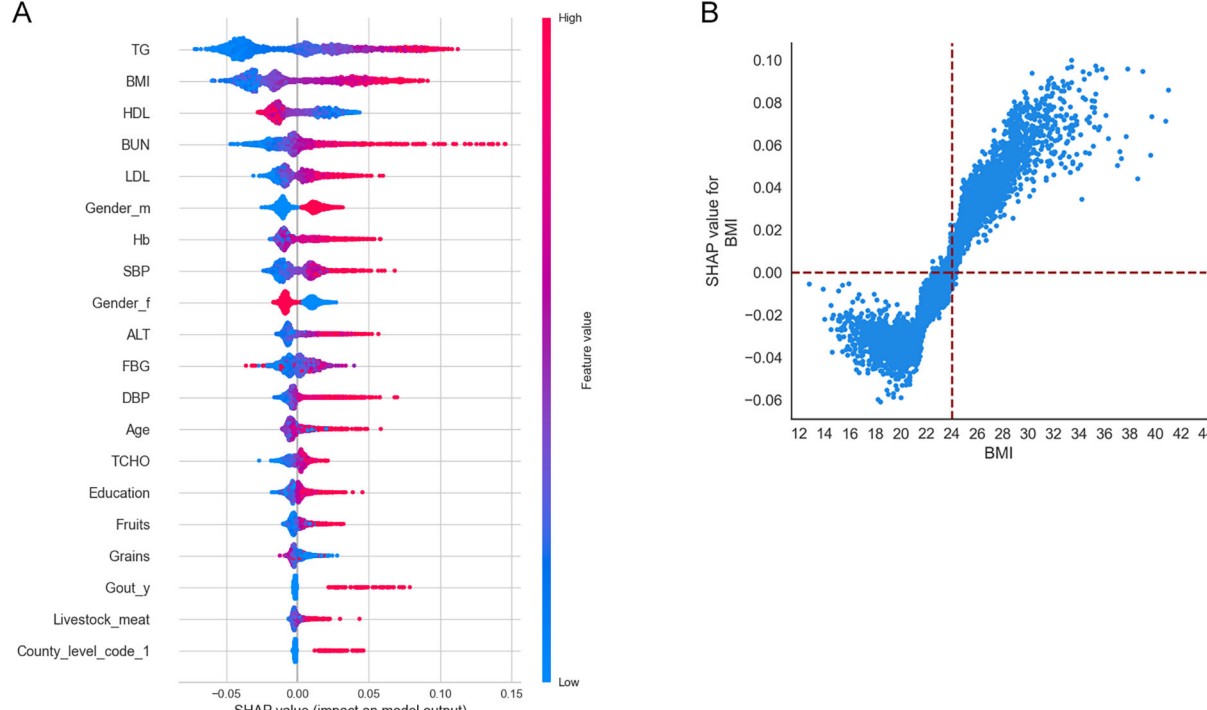

**FIG 1** Association of BMI and HUA, revealed by the interpretable machine learning approach. (A) The summary plot shows the SHAP values of all individuals. The position of the point on the horizontal axis quantifies the effect of a single characteristic on the random forest classifier's prediction for a particular individual. Colors represent the actual value of the features, with blue and red representing low and high. (B) The dependence plot illustrates the trend of the SHAP value for BMI changes with BMI.

0.94%, $P < 0.05$) (Fig. S3). However, *Bacteroidetes*, *Firmicutes,* and *Verrucomicrobia* did not mediate the association of BMI with SUA (Table S4).

Furthermore, we evaluated the mediating effects of gut microbiota diversity at the genus level. The results showed that 25 bacteria could partially mediate the association of BMI with SUA (Fig. 3; Fig. S4 and Table S5). Particularly, 15 bacteria (*Ralstonia*, *f_Coma-monadaceae*, *f_Caulobacteraceae*, *Bradyrhizobium*, *Halomonas*, *Ochrobactrum*, *Pseudidio-marina*, *Odoribacter*, *Delftia*, *Oscillospira*, *Faecalibacterium*, *f_Rikenellaceae*, *f_Methylobacteriaceae*, *f_[Barnesiellaceae]*, *Sediminibacterium*) had mediation effect values greater than 1%. For instance, 2.76% of the total effects of BMI were mediated by *Ralstonia,* and 1.23% of the total effects of BMI were mediated by *Oscillospira* (Fig. 3). To make our results more reliable, we used partial sampling and randomly selected 5,280, 4,280, and 3,280 samples for mediated effects analysis. We found 10 species were better reproduced, such as *Ralstonia*, *Bradyrhizobium*, *Halomonas*, *Ochrobactrum*, *Odoribacter*, *Oscillospira*, etc. (Fig. S5 and Table S6).

**TABLE 2** The associations of BMI with the α-diversity and phylum level of the gut mircobiota[a]

|                    | β      | P     | FDR   | 95% CI           |
|--------------------|--------|-------|-------|------------------|
| PD whole tree      | −0.012 | 0.408 | 0.536 | (−0.042, 0.017)  |
| Observed species   | −0.132 | 0.499 | 0.653 | (−0.513, 0.25)   |
| Chao1              | −0.284 | 0.213 | 0.301 | (−0.732, 0.163)  |
| Shannon            | 0      | 0.98  | 0.98  | (−0.008, 0.007)  |
| *Verrucomicrobia*  | −0.002 | 0     | 0     | (−0.003, −0.001) |
| *Proteobacteria*   | 0.002  | 0.006 | 0.054 | (0.001, 0.003)   |
| *Firmicutes*       | 0.004  | 0     | 0     | (0.003, 0.006)   |
| *Bacteroidetes*    | −0.004 | 0     | 0     | (−0.006, −0.003) |

[a]Model was adjusted for age, gender, TG, BUN, HDL, Hb, LDL, FBG, SBP, DBP, ALT, TCHO, grains, livestock_meat, and fruits.

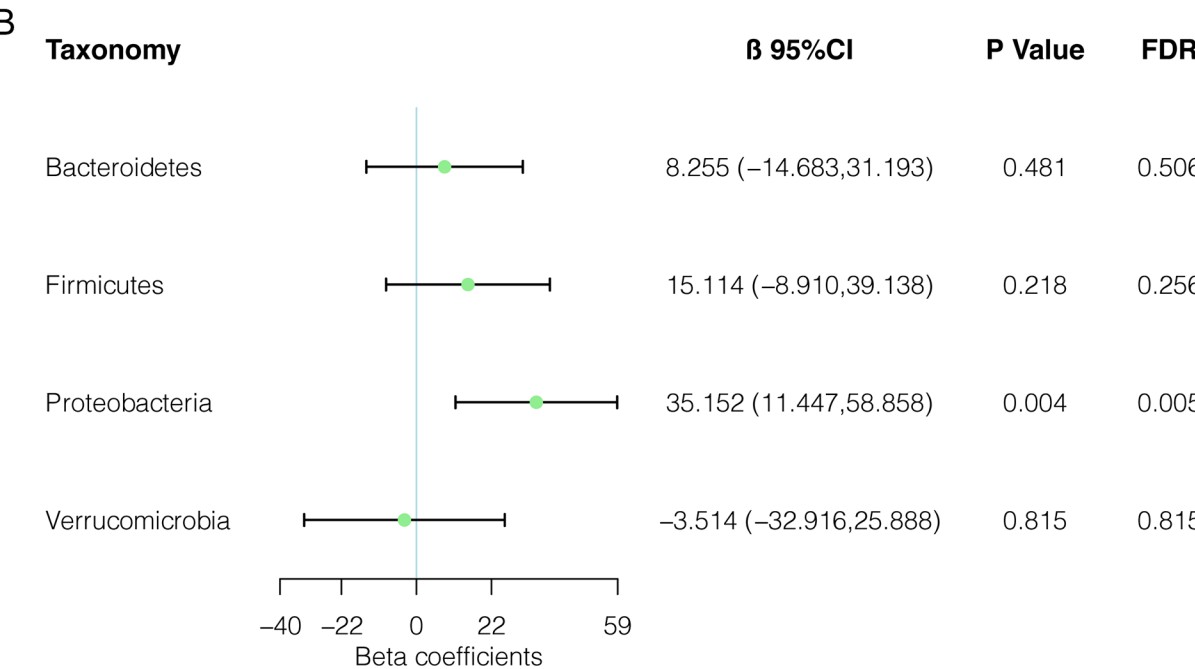

**FIG 2** The association of gut microbiota with SUA. (A) The associations of the α-diversity of the gut microbiota with SUA. (B) The associations of phylum of gut microbiota with SUA. Model was adjusted for age, gender, TG, BUN, HDL, Hb, LDL, FBG, SBP, DBP, ALT, TCHO, grains, livestock_meat, and fruits.

## DISCUSSION

Obesity proceeded to the incidence of hyperuricemia, and BMI was positively correlated with serum uric acid levels in previous studies, which was confirmed by our findings (30, 31). Studies clarified that hyperuricemia in obesity is mainly attributed to the increased intake of purines and an impaired renal clearance of UA (13, 32). Also, xanthine oxidore-ductase activity increased in adipose tissue, which degrades hypoxanthine and xanthine to uric acid (33). Increasing evidence shows that the gut microbiota plays an important

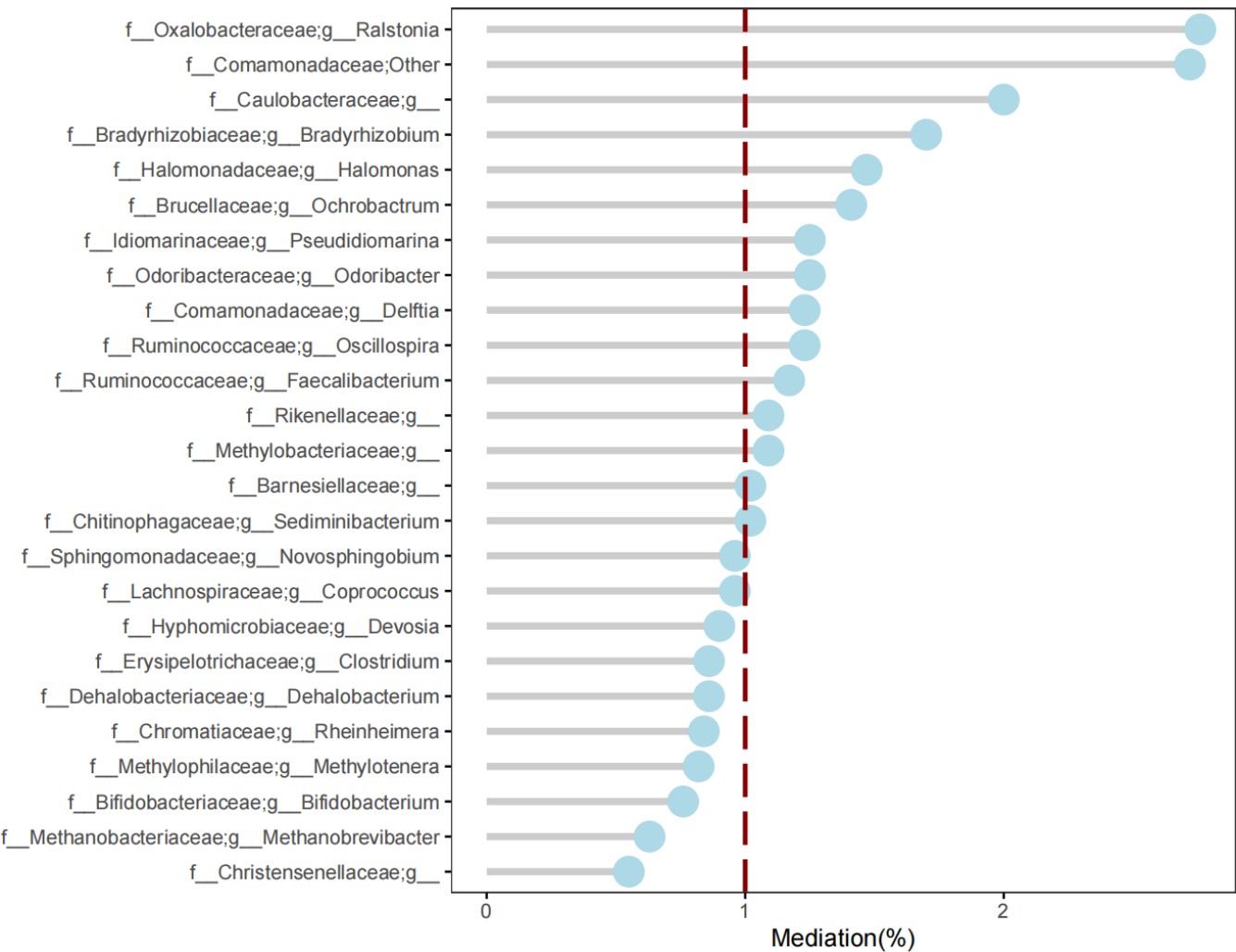

**FIG 3** The mediating effects (%, proportion mediated) of bacterial genus on the associations of BMI with SUA. The mediation analyses were adjusted for age, gender, TG, BUN, HDL, Hb, LDL, FBG, SBP, DBP, ALT, TCHO, grains, livestock_meat, and fruits.

role in the development of obesity and obesity-related complications, such as hyperlipi-demia and diabetes (34). However, little research has been done on the roles of gut microbiota in the development of obesity-related hyperuricemia. In the present study, we first provided evidence supporting the mediator role of gut microbiota characteristics alterations in the association between BMI and SUA.

The association between obesity and human gut microbiota was explored in several studies over the past few years (35–37). For example, studies described that obese individuals exhibited a higher *Firmicutes/Bacteroidetes* ratio, compared to the normal-weight individuals (38). In our assessment of the relationship between BMI and gut microbiota, we found *Proteobacteria* was positively associated with BMI level. Addition-ally, several gut microbes were found to be associated with BMI in recent studies. Deng et al. found that higher BMI was associated with lower abundances of four species (*Clostridium hathewayi*, *Parabacteroides unclassified*, *Lachnospiraceae bacterium 3 1 57FAA CT1*, and *Lachnospiraceae bacterium 7 1 58FAA*) (39). A cross-sectional study among Chinese college students observed that *Anaerotruncus*, *Parabacteroides,* and *Alistipes* were negatively related to BMI, which was consistent with our findings. We also identified that 52 genera were positively associated with BMI, and 50 genera were negatively associated with BMI. For example, *Blautia* and *Ralstonia* were positively associated with BMI, but some butyric acid-producing bacteria, such as *Butyricimonas* and *Oscillospira,* were negatively associated with BMI.

The diversity and abundance of gut microbiota were altered in hyperuricemia. However, due to the high complexity of gut microbiota, affected by many confounding factors such as geography, environment, and diet, how and what gut microbes may be associated with SUA level or hyperuricemia were not clear. Wei et al. demonstrated low relative abundances of genus *Coprococcus* were associated with high levels of serum urate (17), which was confirmed by our findings. In this study, we found α-diversity (Chao1) was negatively associated with SUA level. At the phylum level, the increase in the abundance of *Proteobacteria* was associated with higher SUA. Furthermore, we found 16 genera that related to BMI were also significantly associated with SUA.

The mediation analysis suggested that the association of BMI with SUA was partially mediated by the gut microbiota. The majority of *Proteobacteria* was positively associated with BMI and SUA and mediated 0.94% of the total effects of BMI on SUA. A recent study reported that compared to the controls, *Proteobacteria* was more abundant in tophaceous gout, caused by hyperuricemia (40). Therefore, we hypothesize that *Proteobacteria* may drive the hyperuricemia and play an important role in the development of hyperuricemia. Furthermore, we only used BMI-associated microbes in the next mediation analysis associated between BMI and SUA. We found that 25 genera may partially mediate the association of BMI with SUA. Besides, we can get better reproducibility through partial random sampling. Among them, *Ralstonia* exerts the largest mediating effect in the association of BMI and SUA (2.79%). It has been observed that the fecal abundance of *Ralstonia pickettii* was increased in obese subjects with pre-diabetes and Diabetes mellitus type 2 (T2DM) (41). *Ralstonia pickettii* was also found to facilitate the transfer of lipopolysaccharide to the blood, and it explains the increased low-grade inflammation observed in obese/diabetic patients (42). These further demonstrate that *Ralstonia* may be important for the development of obesity. We also found that *Rikenellaceae* was negatively associated with SUA and mediated the effects of BMI. Previous studies have also revealed that the *Rikenellaceae* family can produce short-chain fatty acid butyrate and helps to maintain good gut condition (43). This suggests that *Rikenellaceae* is beneficial for host metabolism. However, all the results should be identified using the shotgun metagenomic sequencing technology or prospective, large-sample, multicenter studies in the future.

Intestinal microecology may be a new treatment for hyperuricemia by regulating intestinal flora and restoring intestinal microecological homeostasis through microbial agents or fecal bacteria transplantation. But for obese people, although our study suggests that 25 genera exert mediating effects, it is worth noting that these effect sizes are small. Therefore, approaches to prevent or ameliorate hyperuricemia by intervening gut microbiota in obese people remain to be further verified.

## Strengths and limitations

Our study has some strengths. First, this is one of the largest Eastern-population-based gut microbiome data sets, ensuring sufficient statistical power to test the associations among BMI, the gut microbiome, and SUA (22). Second, the large sample size allowed us to adequately adjust for many covariates, including clinical indicators and dietary intake. These covariates are also important factors related to SUA and the gut microbiome.

We admit that our study has some limitations. First, our study is a cross-sectional study; therefore, we cannot describe the causality between BMI, the gut microbiota, and SUA. Second, our study explored the association between BMI and gut microbiota by using data from 16S rRNA sequencing; thus, we cannot detect gut microbes at the species level. It should be further validated whether the identified associations in our cohort are specific in the future larger studies. Finally, the mediation effect size was relatively low. Whether microbiota can be used as a target for prevention/amelioration of hyperuricemia in obese people needs to be further confirmed.

## Conclusion

Our study demonstrated that there was a mediating effect of gut microbiota between BMI and SUA, which suggested the microbiota has an effect on increasing uric acid levels in obese people.

## ACKNOWLEDGMENTS

We thank all participants and staff from the Guangdong Gut Microbiome Project.

This work was supported by the National Natural Science Foundation of China (81800746).

The authors declare that they have no competing interests.

## AUTHOR AFFILIATIONS

[1]Department of Laboratory Medicine, Microbiome Medicine Centre, Zhujiang Hospital, Southern Medical University, Guangzhou, Guangdong, China
[2]Guangdong Provincial Institute of Public Health, Guangdong Provincial Centre for Disease Control and Prevention, Guangzhou, Guangdong, China
[3]Guangdong Provincial Clinical Research Center for Laboratory Medicine, Guangzhou, Guangdong, China

## AUTHOR ORCIDs

Zhuo Duan  http://orcid.org/0009-0000-5984-2542
Jingxiang Fu  http://orcid.org/0000-0003-3009-5269

## AUTHOR CONTRIBUTIONS

Zhuo Duan, Data curation, Writing – original draft | Jingxiang Fu, Data curation, Writing – original draft | Feng Zhang, Writing – original draft | Yijia Cai, Writing – review and editing | Guangyan Wu, Writing – review and editing | Wenjun Ma, Data curation | Hongwei Zhou, Writing – review and editing | Yan He, Writing – review and editing

## DATA AVAILABILITY

The raw data for 16S rRNA gene sequences are available in NCBI under accession number PRJEB18535.

## ADDITIONAL FILES

The following material is available online.

### Supplemental Material

**Supplemental material (Spectrum01140-23-s0001.docx).** Supplemental figures and tables.

### Open Peer Review

**PEER REVIEW HISTORY (review-history.pdf).** An accounting of the reviewer comments and feedback.

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
