## [Reviewer comments · Microbiology Spectrum]

Microbiology Spectrum

The association between BMI and serum uric acid is partially mediated by gut microbiota.

Zhuo Duan, Jingxiang Fu, Feng Zhang, Yijia Cai, Guangyan Wu, Wenjun Ma, Hong-Wei Zhou, and Yan He

Corresponding Author(s): Yan He, Southern Medical University

Review Timeline:

Submission Date:	March 15, 2023
Editorial Decision:	June 5, 2023
Revision Received:	June 26, 2023
Accepted:	July 20, 2023

Editor: Wei-Hua Chen

Reviewer(s): Disclosure of reviewer identity is with reference to reviewer comments included in decision letter(s). The following individuals involved in review of your submission have agreed to reveal their identity: Xiaoyu Gao (Reviewer #1); Khemlal Nirmalkar (Reviewer #3)

Transaction Report:

DOI: <https://doi.org/10.1128/spectrum.01140-23>

June 5, 2023

Dr. Yan He
Southern Medical University
Guangzhou
China

Re: Spectrum01140-23 (**The association between BMI and serum uric acid is partially mediated by gut microbiota**)

Dear Dr. Yan He:

Thank you for submitting your manuscript to Microbiology Spectrum. After careful evaluation by three external experts, we have determined that your paper shows promise. However, in order to proceed with publication, significant revisions are required.

Please address the concerns raised by the reviewers and submit a revised version of your study. Their feedback will help strengthen your research and improve its impact in the field of microbiology.

We appreciate your commitment to this research and look forward to receiving your revised manuscript. Thank you for choosing Microbiology Spectrum as a platform for your work.

Link Not Available

Sincerely,

Wei-Hua Chen

Journals Department
Reviewer comments:

Reviewer #1 (Comments for the Author):

It is my honor to review this manuscript by Zhuo et al (Spectrum01140-23). This is a typical cross-sectional study from the GGMP program in Guangdong, China. Just as the author mentioned in the manuscript, there have been many studies on the relationship between obesity (BMI) and hyperuricemia (uric acid), and there are also many studies on the relationship between

gut microbiota (GM) and obesity, GM and hyperuricemia. However, it is not clear whether GM is involved in the pathway mediating the effect of obesity on hyperuricemia. Therefore, this study provides an important clue to further clarify the pathogenesis of elevated SUA levels.

Possible problems:

1. What is the relative abundance of some important microbial groups mentioned in the manuscript, such as Proteobacteria, Ralstonia, Oscillospira, Rikenellaceae and other genus-level microbes (mentioned in the abstract)? No statistical support is given in the text. Is it possible that there is a statistical level of difference?
2. Through data analysis, literature research and comprehensive discussion, the author tried to reveal the possible role of intestinal flora in the way obesity affects hyperuricemia, but I think these correlation analyses based on 16S results may have certain risks. Therefore, the q-PCR should be used to quantify some important microbial groups, and then analyze the correlation to further confirm the rationality of these inferences. Of course, this suggestion may be difficult for the author to implement. A random partial sample might be a good idea.
3. p29: "genus", p224: "genera", but many non-genus level microbes are described, such as p30 "Rikenellaceae". So, try to examine these in detail.
4. p133--134: Reference [24], which does not look like a paper by the author (including all the named authors). So, "our" should be deleted.
5. I cannot find the position of Fig S4 in the main text of the manuscript.

Reviewer #2 (Comments for the Author):

This is a large and interesting study from the Guangdong Gut Microbiome Project that investigates the relationship between the gut microbiome, obesity, and serum uric acid (SUA), a marker of hyperuricemia. Strengths include the large sample size, ample metadata, and inclusion of a large non-Western cohort that was robustly recruited from 14 districts in Guangdong Province, China. Over 6000 participants had 16S rRNA gene sequence data, serum SUA, and BMI information and were included in this study. The authors performed three main analyses to associate gut microbiome features to BMI, SUA, and to link BMI-SUA-genera. They first used a random forest machine learning classifier to predict clinical characteristics and used a program, SHAP, to infer directionality of feature contributions. They show, using this approach, that high BMI is linked with a higher probability for hyperuricemia. Next, they ran analyses to correlate bacterial features (phyla and genera) with BMI and then SUA. Finally, they performed an interesting and innovative mediation analysis to link BMI-SUA and gut microbial features. Overall, the study was well powered and exciting, but was underdeveloped.

Comments:

High BMI has already been shown to be a risk factor for hyperuricemia, so the novelty of Fig. 1 is unclear. I'd be more interested in running Random Forest to predict SUA or high BMI based on gut microbiome features.

The group has access to so much interesting metadata, but the analyses feels underdeveloped in incorporating variables outside of BMI and SUA in the context of the gut microbiota.

The phylum level analyses are difficult to interpret, since this is such a high taxonomic level. The genus level findings were informative and exciting. I'd recommend highlighting the genus level results.

It would be helpful to confirm the taxonomic classification of Ralstonia - this genus is most commonly associated with plants (but is related to Pseudomonas, so it's possible this was misclassified. It may be a true finding, but I would like to see this confirmed). Another approach might be to attempt to identify the feature to species level and compare to R. pickettii.

The figures are exciting but are not well described in the Results section. I'd recommend expanding the results.

Figure S4 is not referenced in the Results section

The 's' in 16S needs to be capitalized

Reviewer #3 (Public repository details (Required)):

Raw sequences submission should be mentioned in the text in NCBI or any other appropriate repository.

Reviewer #3 (Comments for the Author):

Comments:

1. why did the authors use qiime1? Qiime2 has been available for a long time which is better to identify microbiota via amplicon

sequence variant instead of clustering the OTUs.

2. what was the database and OTUs threshold used for taxonomic identification? It should be mentioned in the method section.
3. It is confusing whether SUA and BMI association/mediation analysis was performed with microbe for all participants or only with HUA.
4. Did the author check the microbial difference between NOR and HUA groups?
5. HUA represents only 23% out of 6280, mediation analyses should be performed separately.
6. Previous reports have shown some different bacteria than this current study. Can the author explain, why different microbes are associated with SUA and BMI? Does it matter about geographical location or population? It should be part of the discussion.
7. Diet should be discussed, as diet plays an important role in obesity/BMI and microbiota composition.
8. Raw sequences submission should be mentioned in the text in NCBI or any other appropriate repository.

Staff Comments:

Preparing Revision Guidelines

Please return the manuscript within 60 days; if you cannot complete the modification within this time period, please contact me. If you do not wish to modify the manuscript and prefer to submit it to another journal, please notify me of your decision immediately so that the manuscript may be formally withdrawn from consideration by Microbiology Spectrum.

It is my honor to review this manuscript by Zhuo et al (Spectrum01140-23). This is a typical cross-sectional study from the GGMP program in Guangdong, China. Just as the author mentioned in the manuscript, there have been many studies on the relationship between obesity (BMI) and hyperuricemia (uric acid), and there are also many studies on the relationship between gut microbiota (GM) and obesity, GM and hyperuricemia. However, it is not clear whether GM is involved in the pathway mediating the effect of obesity on hyperuricemia. Therefore, this study provides an important clue to further clarify the pathogenesis of elevated SUA levels.

Possible problems:

1. What is the relative abundance of some important microbial groups mentioned in the manuscript, such as Proteobacteria, Ralstonia, Oscillospira, Rikenellaceae and other genus-level microbes (mentioned in the abstract) ? No statistical support is given in the text. Is it possible that there is a statistical level of difference?
2. Through data analysis, literature research and comprehensive discussion, the author tried to reveal the possible role of intestinal flora in the way obesity affects hyperuricemia, but I think these correlation analyses based on 16S results may have certain risks. Therefore, the q-PCR should be used to quantify some important microbial groups, and then analyze the correlation to further confirm the rationality of these inferences. Of course, this suggestion may be difficult for the author to implement. A random partial sample might be a good idea.
3. p29: "genus", p224: "genera", but many non-genus level microbes are described, such as p30 "Rikenellaceae". So, try to examine these in detail.
4. p133--134: Reference [24], which does not look like a paper by the author (including all the named authors). So, "our" should be deleted.
5. I cannot find the position of Fig S4 in the main text of the manuscript

Dear Editor,

Thanks very much for taking your time to review this manuscript. We appreciate editor and reviewers for their positive and constructive comments and suggestions on our manuscript entitled “The association between BMI and serum uric acid is partially mediated by gut microbiota”(ID: Spectrum01140-23). We have studied the comments carefully and have made revision in this paper. Our point-by-point responses are provided below. Our responses are written in blue for clarity.

Reviewer #1 (Comments for the Author):

1. What is the relative abundance of some important microbial groups mentioned in the manuscript, such as Proteobacteria, Ralstonia, Oscillospira, Rikenellaceae and other genus-level microbes (mentioned in the abstract) ? No statistical support is given in the text. Is it possible that there is a statistical level of difference?

Response 1

Thank you for your comment. We have supplemented the relative abundance, the mediation effects and the P value of these important microbial as follows (Table for review 1). The mediation effects and P value are also shown in TableS5 in the Supplementary materials.

Table for review 1.

Taxon	Level	Relative abundance (mean±SD)	Mediation (%)	95%CI	P
g__Odoribacter	Genus	0.000732±0.001522	1.25	(0.026, 0.112)	0.005
f__Christensenellaceae;g__	Genus	0.001109±0.00556	0.55	(0.005, 0.053)	0.037
g__Dehalobacterium	Genus	0.000134±0.000567	0.86	(0.014, 0.073)	0.01
g__Oscillospira	Genus	0.019668±0.024632	1.23	(0.012, 0.112)	0.015
g__Ralstonia	Genus	0.018851±0.033692	2.76	(0.069, 0.203)	<0.001
g__Bradyrhizobium	Genus	0.002553±0.004592	1.7	(0.028, 0.149)	0.01
f__Caulobacteraceae;g__	Genus	0.003419±0.006061	2	(0.040, 0.165)	0.002
g__Faecalibacterium	Genus	0.076136±0.082788	1.17	(-0.116, -0.010)	0.028
g__Ochrobactrum	Genus	0.00045±0.000815	1.41	(0.024, 0.129)	0.009
g__Clostridium	Genus	0.001045±0.005234	0.86	(-0.071, -0.017)	0.002
f__Methylobacteriaceae	Genus	0.000068±0.000256	1.09	(0.019, 0.090)	0.005
g__Sediminibacterium	Genus	0.014571±	1.02	(0.006, 0.100)	0.039

			0.029652			
f__Rikenellaceae;g__	Genus	0.016486± 0.027659	1.09	(0.014, 0.098)	0.014	
f__Comamonadaceae;Other	Genus	0.000558± 0.001424	2.72	(0.072, 0.203)	<0.001	
g__Coproccoccus	Genus	0.010923± 0.019577	0.96	(-0.094, -0.011)	0.03	
g__Devosia	Genus	0.000111± 0.000271	0.9	(0.011, 0.090)	0.037	
g__Pseudidiomarina	Genus	0.000092± 0.000234	1.25	(0.019, 0.113)	0.01	
f__[Barnesiellaceae];g__	Genus	0.0016±0.004575	1.02	(0.020, 0.088)	0.005	
g__Rheinheimera	Genus	0.000039± 0.000155	0.84	(0.007, 0.087)	0.049	
g__Halomonas	Genus	0.000158± 0.000351	1.47	(0.025, 0.123)	0.005	
g__Delftia	Genus	0.000081± 0.000265	1.23	(0.021, 0.111)	0.012	
g__Methanobrevibacter	Genus	0.001681± 0.008421	0.63	(0.007, 0.064)	0.031	
g__Methylothera	Genus	0.000013± 0.000071	0.82	(0.008, 0.083)	0.047	
g__Bifidobacterium	Genus	0.005514± 0.020581	0.76	(-0.071, -0.007)	0.026	
g__Novosphingobium	Genus	0.000045± 0.000161	0.96	(0.010, 0.097)	0.033	

2. Through data analysis, literature research and comprehensive discussion, the author tried to reveal the possible role of intestinal flora in the way obesity affects hyperuricemia, but I think these correlation analyses based on 16S results may have certain risks. Therefore, the q-PCR should be used to quantify some important microbial groups, and then analyze the correlation to further confirm the rationality of these inferences. Of course, this suggestion may be difficult for the author to implement. A random partial sample might be a good idea.

Response 2

Thank you for your comment. We used partial sampling and randomly selected three times including 5280, 4280 and 3280 samples for mediated effects analysis, among which 10 species were better reproduced, such as *Ralstonia*, *Bradyrhizobium*,

Halomonas, *Ochrobactrum*, *Odoribacter*, *Oscillospira*, etc. We have added this part of the results (p247): “To make our results more reliable, we used partial sampling and randomly selected 5280, 4280 and 3280 samples for mediated effects analysis. We found 10 species were better reproduced, such as *Ralstonia*, *Bradyrhizobium*, *Halomonas*, *Ochrobactrum*, *Odoribacter*, *Oscillospira*, etc (Fig S5 and Table S6).” in the mediation effects of our manuscript. Of course, as you mentioned above, it is a better choice to validate our results by using qPCR. Unfortunately, as the samples for our study were collected many years ago, the sample conditions were not sufficient for additional sequencing analysis. However, in our future studies, metagenomic sequencing will be adopted for sure.

Fig S5. Significant mediated effect genus in random partial sampling groups. The Venn plot illustrates the number and percent of the overlapping genus in different sample size groups.

Table S6. Significant mediated effect genus reproduced in the partially random sampling analyses.

Taxon	6280	5280	4280	3280
g__Odoribacter	1	1	1	1
f__Christensenellaceae;g__	1	1	1	1
g__Dehalobacterium	1	1	1	1
g__Oscillospira	1	1	1	1
g__Ralstonia	1	1	1	1
g__Bradyrhizobium	1	1	1	1
f__Caulobacteraceae;g__	1	1	1	1
g__Faecalibacterium	1	1	1	0

g__Ochrobactrum	1	1	1	1
g__Clostridium	1	1	1	0
f__Methylobacteriaceae	1	0	0	0
g__Sediminibacterium	1	1	1	0
f__Rikenellaceae;g__	1	1	1	1
f__Comamonadaceae;Other	1	0	0	0
g__Coproccoccus	1	1	1	1
g__Devosia	1	0	0	0
g__Pseudidiomarina	1	0	0	0
f__[Barnesiellaceae];g__	1	0	0	0
g__Rheinheimera	1	1	0	0
g__Halomonas	1	1	0	0
g__Delftia	1	0	0	0
g__Methanobrevibacter	1	0	0	0
g__Methylothera	1	0	0	0
g__Bifidobacterium	1	0	1	0
g__Novosphingobium	1	0	0	0

3. p29: "genus", p224: "genera", but many non-genus level microbes are described, such as p30 "Rikenellaceae". So, try to examine these in detail.

Response 3

Thank you for pointing out these errors. We have carefully checked the manuscript and corrected the errors.

4. p133--134: Reference [24], which does not look like a paper by the author (including all the named authors). So, "our" should be deleted.

Response 4

Thank you very much for pointing this out. We have checked the literature carefully and corrected it in the revised manuscript. (p131)

5. I cannot find the position of Fig S4 in the main text of the manuscript.

Response 5

Thank you for reminding us to clarify this detail. We have corrected it in the revised manuscript. (p241)

Reviewer #2 (Comments for the Author):

1.High BMI has already been shown to be a risk factor for hyperuricemia, so the novelty of Fig. 1 is unclear. I'd be more interested in running Random Forest to predict SUA or high BMI based on gut microbiome features.

Response 1

Thank you for your constructive comment. However, the focus of this paper is aim to investigate the mediating effects of gut microbial dysbiosis in the association between BMI with SUA, rather than to develop a diagnostic model. We will consider conducting analytical studies in this area. Although high BMI has been reported as a risk factor for hyperuricemia, the SHAP analysis was used in this study Fig1 to better illustrate the linear relationship between changes in BMI and the risk of developing HUA, using the data from this cohort to supplement the evidence of this phenomenon and better illustrate the high impact of BMI on developing HUA among many factors.

2.The group has access to so much interesting metadata, but the analyses feels underdeveloped in incorporating variables outside of BMI and SUA in the context of the gut microbiota.

Response 2

GGMP data have been used in several excellent articles^[1-5]. In this paper, we focused on providing evidence for the associations among BMI, gut microbiota, and SUA to investigate whether alterations in the gut microbiota lie in the pathways from obesity to hyperuricemia. It is because GGMP has many interesting metadata that we can adjust for multiple confounding factors in our analysis and make our results more reliable. And we choose the top 15 baseline characteristics associated with risks of HUA as covariates in the data analyses: Age, Gender, TG, BUN, HDL, Hb, LDL, FBG, SBP, DBP, ALT, TCHO, Grains, Livestock_meat, Fruits. We have added these information: “The top 15 baseline characteristics associated with risks of HUA were chose as covariates in all correlation and mediation analyses: Age, Gender, TG (Triglyceride), BUN (Blood Urea Nitrogen), HDL (High Density Lipoprotein), Hb

(Hemoglobin), LDL (Low Density Lipoprotein), FBG (Fasting Blood Glucose), SBP (Systolic Blood Pressure), DBP (Diastolic Blood Pressure), ALT (Alaninetransaminase), TCHO (Total Cholesterol), Grains, Livestock_meat, Fruits. ” in Materials and methods. (p162)

3.The phylum level analyses are difficult to interpret, since this is such a high taxonomic level. The genus level findings were informative and exciting. I'd recommend highlighting the genus level results.

Response 3

Thank you for your comment. We agree with the reviewer's comments and have given more prominence to the genus-level results. For example, we have added a random partial sample analyse in the mediation results to observe the reproducibility of these genera.(p247)

4.It would be helpful to confirm the taxonomic classification of *Ralstonia* - this genus is most commonly associated with plants (but is related to *Pseudomonas*, so it's possible this was misclassified. It may be a true finding, but I would like to see this confirmed). Another approach might be to attempt to identify the feature to species level and compare to *R. pickettii*.

Response 4

We agree with you that *Ralstonia* is most commonly associated with plants and related to *Pseudomonas*. However, we think the process of collecting and processing stool samples was relatively rigorous, described in our earlier paper^[1]. Besides, our sequencing results were able to distinguish between *Ralstonia* and *Pseudomonas* as follows (Figure for review1). As 16S rRNA sequencing cannot reach species and strain-level taxonomic classification, we cannot identify the feature to species level and compare it to *R. pickettii*. Future studies employing shotgun metagenomic sequencing technology will be needed to fill this gap.

A	
ID	
k__Bacteria;p__Bacteroidetes;c__Bacteroidia;o__Bacteroidales;f__Porphyromonadaceae;g__Parabacteroides	
k__Bacteria;p__Bacteroidetes;c__Bacteroidia;o__Bacteroidales;f__[Odoribacteraceae];g__Butyricimonas	
k__Bacteria;p__Firmicutes;c__Clostridia;o__Clostridiales;f__Ruminococcaceae;g__Anaerotruncus	
k__Bacteria;p__Bacteroidetes;c__Bacteroidia;o__Bacteroidales;f__Bacteroidaceae;g__Bacteroides	
k__Bacteria;p__Bacteroidetes;c__Bacteroidia;o__Bacteroidales;f__[Odoribacteraceae];g__Odoribacter	
k__Bacteria;p__Firmicutes;c__Clostridia;o__Clostridiales;f__Christensenellaceae;g__Christensenella	
k__Bacteria;p__Firmicutes;c__Clostridia;o__Clostridiales;f__Christensenellaceae;g__	
k__Bacteria;p__Firmicutes;c__Clostridia;o__Clostridiales;f__Lachnospiraceae;g__Clostridium	
k__Bacteria;p__Firmicutes;c__Clostridia;o__Clostridiales;f__Lachnospiraceae;g__Blautia	
k__Bacteria;p__Firmicutes;c__Clostridia;o__Clostridiales;f__Dehalobacteriaceae;g__Dehalobacterium	
k__Bacteria;p__Firmicutes;c__Clostridia;o__Clostridiales;f__Ruminococcaceae;g__Oscillospira	
k__Bacteria;p__Proteobacteria;c__Betaproteobacteria;o__Burkholderiales;f__Oxalobacteraceae;g__Ralstonia	
k__Bacteria;p__Proteobacteria;c__Alphaproteobacteria;o__Rhizobiales;f__Bradyrhizobiaceae;g__Bradyrhizobium	
k__Bacteria;p__Firmicutes;c__Clostridia;o__Clostridiales;f__[Tissierellaceae];g__WAI_1855D	
k__Bacteria;p__Proteobacteria;c__Gammaproteobacteria;o__Pseudomonadales;f__Pseudomonadaceae;g__Pseudomonas	
k__Bacteria;p__Proteobacteria;c__Betaproteobacteria;o__Burkholderiales;f__Alcaligenaceae;g__Sutterella	

Figure for review 1. Part of the OTU table used in this study

5.The figures are exciting but are not well described in the Results section. I'd recommend expanding the results.

Response 5

Thank you for your comment. We have added : “Next, we only chose BMI-associated genera in the analysis of the association between SUA and gut microbes.” and “To make our results more reliable, we used partial sampling and randomly selected 5280, 4280 and 3280 samples for mediated effects analysis. We found 10 species were better reproduced, such as *Ralstonia*, *Bradyrhizobium*, *Halomonas*, *Ochrobactrum*, *Odoribacter*, *Oscillospira*, etc (Fig S5 and Table S6).” in the Results section.(p225 and p247)

6.Figure S4 is not referenced in the Results section

Response6

We have corrected it in the revised manuscript. (p 241)

7.The 's' in 16S needs to be capitalized

Response7

We have checked the literature carefully and corrected it in the revised manuscript.

Reviewer #3 (Comments for the Author):

1. why did the authors use qiime1? Qiime2 has been available for a long time which is better to identify microbiota via amplicon sequence variant instead of clustering the OTUs.

Response1

Thank you for your comment. We apologize for any confusion this may cause you. Our group have been doing amplicon sequencing for many years, though some commands of our established pipeline are still from qiime1, our analysis pipeline has been updated to identify microbiota via amplicon sequence variant methods (e.g., DADA2, Deblur) rather than clustering OTUs. Besides, we also add the detail of data preprocessing in Response2 below and the “Materials and methods” part of our manuscript.

2. what was the database and OTUs threshold used for taxonomic identification? It should be mentioned in the method section.

Response2

The preprocessing analysis details are implemented in the “Materials and methods” section(p134), also as follows:

Representative sequences were aligned using the PyNAST^[6] algorithm. The Ribosomal Database Project classifier^[7] against the Greengenes database^[8] (version 13.8) was used to perform taxonomic profiling.

3. It is confusing whether SUA and BMI association/mediation analysis was performed with microbe for all participants or only with HUA.

Response3

Thank you for reminding us to clarify this detail. The association/mediation analysis was observed in all participants. We have added this information in our “Materials and methods” section.(p161)

4. Did the author check the microbial difference between NOR and HUA groups?

Response4

Compared with participants with Nor, a significant reduction was observed in α diversity of intestinal microbiota from the HUA group, suggesting that a lower microbial diversity in the intestine was associated with HUA (Figure for review 2A). Moreover, PCoA analysis plot constructed by Bray-curtis distances also showed the composition and abundance of the microbiota in HUA were significantly different from those of the microbiota in Nor (Figure for review 2B, as follows). Meanwhile, the composition of intestinal microbiota in HUA group was different from that in Nor group (Figure for review 2C). Analysis with the linear discriminant analysis (LDA) effect size (LEfSe) method showed that the relative abundance of Proteobacteria, Burkholderiales, Enterobacteriales, Oxalobacteraceae increased, while the relative abundance of Bacteroidales, Ruminococcaceae, Prevotella decreased in the HUA group (Figure for review 2D, as follows). However, the focus of this paper is to investigate the mediating effects of gut microbial dysbiosis in the association between BMI with SUA. So, we did not include this part of the results in our manuscript.

Figure for review 2. Characteristics of intestinal flora in hyperuricemia population.

5. HUA represents only 23% out of 6280, mediation analyses should be performed separately.

Response5

Thank you for your comment. However, our study was to explore whether BMI-associated gut microbiota mediates elevated uric acid levels. Not only high uric acid value but also low uric acid value are needed. Therefore, we believe that the mediation effect analysis should be carried out in the whole population. Of course, we also supplemented the results of partial sampling to make our analysis results more reliable.

6. Previous reports have shown some different bacteria than this current study. Can the author explain, why different microbes are associated with SUA and BMI? Does it matter about geographical location or population? It should be part of the discussion.

Response6

As suggested by the reviewer, we have added: “However, due to the high complexity of gut microbiota, affected by many confounding factors such as geography, environment and diet, how and what gut microbes may be associated with SUA level or hyperuricemia were not clear.” in the Discussion section. (p283)

7. Diet should be discussed, as diet plays an important role in obesity/BMI and microbiota composition.

Response7

We agree with you that diet does have an effect on BMI and SUA. However, this study is a cross-sectional study, which can only do correlation research and cannot demonstrate cause and effect. In spite of this, we also corrected for some dietary factors when conducting correlation analysis and mediation effect analysis. Such as Grains, Livestock_meat and Fruits. A prospective, large-sample, multicenter study is needed to confirm our conclusion in this study in the future.

8. Raw sequences submission should be mentioned in the text in NCBI or any other

appropriate repository.

Response8

We have added it in the “Data Availability” section: “The raw data for 16 S rRNA gene sequences could be found in the European Nucleotide Archive (<https://www.ebi.ac.uk/ena/>) under accession number PRJEB18535.”(p350)

References

- [1] He Y, Wu W, Zheng HM, Li P, McDonald D, Sheng HF, Chen MX, Chen ZH, Ji GY, Zheng ZD, Mujagond P, Chen XJ, Rong ZH, Chen P, Lyu LY, Wang X, Wu CB, Yu N, Xu YJ, Yin J, Raes J, Knight R, Ma WJ, Zhou HW. 2018. Regional variation limits applications of healthy gut microbiome reference ranges and disease models. *Nat Med* 24:1532-1535. <http://dx.doi.org/10.1038/s41591-018-0164-x>.
- [2] Gou W, Ling CW, He Y, Jiang Z, Fu Y, Xu F, Miao Z, Sun TY, Lin JS, Zhu HL, Zhou H, Chen YM, Zheng JS. 2021. Interpretable Machine Learning Framework Reveals Robust Gut Microbiome Features Associated With Type 2 Diabetes. *Diabetes Care* 44(2):358-366. <https://doi.org/10.2337/dc20-1536>.
- [3] Jiang Z, Zhuo LB, He Y, Fu Y, Shen L, Xu F, Gou W, Miao Z, Shuai M, Liang Y, Xiao C, Liang X, Tian Y, Wang J, Tang J, Deng K, Zhou H, Chen YM, Zheng JS. 2022. The gut microbiota-bile acid axis links the positive association between chronic insomnia and cardiometabolic diseases. *Nature communications* 13(1):3002. <https://doi.org/10.1038/s41467-022-30712-x>.
- [4] He Y, Wu W, Wu S, Zheng HM, Li P, Sheng HF, Chen MX, Chen ZH, Ji GY, Zheng ZD, Mujagond P, Chen XJ, Rong ZH, Chen P, Lyu LY, Wang X, Xu JB, Wu CB, Yu N, Xu YJ, Yin J, Raes J, Ma WJ, Zhou HW. 2018. Linking gut microbiota, metabolic syndrome and economic status based on a population-level analysis. *Microbiome* 6(1):172. <https://doi.org/10.1186/s40168-018-0557-6>.
- [5] Jiang Z, Sun TY, He Y, Gou W, Zuo LS, Fu Y, Miao Z, Shuai M, Xu F, Xiao C, Liang Y, Wang J, Xu Y, Jing LP, Ling W, Zhou H, Chen YM, Zheng JS. 2020. Dietary fruit and vegetable intake, gut microbiota, and type 2 diabetes: results from two large human cohort studies. *BMC medicine* 18(1):371. <https://doi.org/10.1186/s12916-020-01842-0>.

[6] Caporaso, J.G., Bittinger, K., Bushman, F.D., DeSantis, T.Z., Andersen, G.L., and Knight, R. (2010). PyNAST: a flexible tool for aligning sequences to a template alignment. *Bioinform. Oxf. Engl.* 26, 266 – 267. <https://doi.org/10.1093/bioinformatics/btp636>.

[7] Wang, Q., Garrity, G.M., Tiedje, J.M., and Cole, J.R. (2007). Naive Bayesian classifier for rapid assignment of rRNA sequences into the new bacterial taxonomy. *Appl. Environ. Microbiol.* 73, 5261 – 5267. <https://doi.org/10.1128/AEM.00062-07>.

[8] DeSantis, T.Z., Hugenholtz, P., Larsen, N., Rojas, M., Brodie, E.L., Keller, K., Huber, T., Dalevi, D., Hu, P., and Andersen, G.L. (2006). Greengenes, a chimera-checked 16S rRNA gene database and workbench compatible with ARB. *Appl. Environ. Microbiol.* 72, 5069 – 5072. <https://doi.org/10.1128/AEM.03006-05>.

We sincerely appreciate the excellent comments provided by the three referees, which have helped to strengthen our paper. We hope that we have adequately responded to the referees regarding the article, and look forward to your evaluation of our revised manuscript. We would be more than happy to make any further changes that will improve the paper and/or facilitate successful publication.

Sincerely,

Zhuo Duan

July 20, 2023

Dr. Yan He
Southern Medical University
Guangzhou
China

Re: Spectrum01140-23R1 (**The association between BMI and serum uric acid is partially mediated by gut microbiota**)

Dear Dr. Yan He:

Your manuscript has been accepted, and I am forwarding it to the ASM Journals Department for publication. You will be notified when your proofs are ready to be viewed.

Sincerely,

Wei-Hua Chen
Editor, Microbiology Spectrum
